# Enhanced-Efficiency Fertilizers Impact on Nitrogen Use Efficiency and Nitrous Oxide Emissions from an Open-Field Vegetable System in North China

**DOI:** 10.3390/plants12010081

**Published:** 2022-12-23

**Authors:** Daijia Fan, Wentian He, Rong Jiang, Daping Song, Guoyuan Zou, Yanhua Chen, Bing Cao, Jiachen Wang, Xuexia Wang

**Affiliations:** Institute of Plant Nutrition, Resources and Environment, Beijing Academy of Agriculture and Forestry Sciences, Beijing 100097, China

**Keywords:** controlled-release urea, nitrification inhibitor, nitrous oxide emission, open vegetable field, cost–benefit analysis

## Abstract

Open vegetable fields in China are a major anthropogenic source of nitrous oxide (N_2_O) emissions due to excessive nitrogen (N) fertilization. A 4 yr lettuce experiment was conducted to determine the impacts of controlled-release fertilizers (CRFs) and nitrification inhibitors (NIs) on lettuce yield, N_2_O emissions and net economic benefits. Five treatments included (i) no N fertilizer (CK), (ii) conventional urea at 255 kg N ha^–1^ based on farmers’ practice (FP), (iii) conventional urea at 204 kg N ha^–1^ (OPT), (iv) CRF at 204 kg N ha^–1^ (CU) and (v) CRF (204 kg N ha^–1^) added with NI (CUNI). No significant differences were found in the lettuce yields among different N fertilization treatments. Compared with FP, the cumulative N_2_O emissions were significantly decreased by 8.1%, 38.0% and 42.6% under OPT, CU and CUNI, respectively. Meanwhile, the net benefits of OPT, CU and CUNI were improved by USD 281, USD 871 and USD 1024 ha^–1^ compared to CN, respectively. This study recommends the combined application of CRF and NI at a reduced N rate as the optimal N fertilizer management for the sustainable production of vegetables in China with the lowest environmental risks and the greatest economic benefits.

## 1. Introduction

Nitrous oxide (N_2_O) is a potent greenhouse gas (GHG) and the greatest contributor to ozone depletion in the stratosphere [1,2]. The atmospheric N_2_O concentration has increased by 22.6% since the preindustrial period as a result of anthropogenic activities, including agricultural production [3]. With the continuing growth of the human population, the higher demand for food requires a substantial input of fertilizer nitrogen (N) into the cropland soils and, hence, makes croplands the largest anthropogenic source of atmospheric N_2_O in recent decades [4]. Thus, there exists an urgent need to develop optimized fertilization strategies that can mitigate N_2_O emissions without detrimental impacts on crop yields [5].

Vegetables are indispensable in the human diet. However, the current vegetable production generally involves intensive fertilization with high N loss potential, which accounts for 9% of the global cropland N_2_O emissions [6,7]. The vegetable cultivation in China has been rapidly expanding due to abundant economic benefits and increasing demand for vegetables, placing China as the leading vegetable producer worldwide, with 51% of the global total production [8]. Compared with cereal crops, higher levels of fertilizer N are applied to Chinese vegetable fields to support vegetables’ rapid growth and achieve high yields, resulting in a greater potential for N losses, including N_2_O emissions [9,10]. Recent study revealed that Chinese vegetable production uses 7.8% of the global synthetic N fertilizers and generates 6.6% of the cropland GHG emissions worldwide [11]. The use of synthetic N fertilizers in the vegetable fields in China caused 33% of the national cropland N_2_O emissions [12].

Open-field cultivation is the dominant vegetable production system in China, comprising 80% and 65% of the total vegetable planting area and output, respectively [13]. The current average N input of Chinese open-field vegetable production is 2.7 times that of the recommended N application rate [14]. Consequently, the open vegetable fields in China remain a hotspot of N_2_O emissions. Differing from greenhouse vegetable production, open vegetable fields are more susceptible to the changes in climatic factors, including air temperature and precipitation, thus resulting in a more complex interplay between soil N transformation and agricultural management practices coupled with wide-ranging weather conditions [15]. Therefore, exploring the responses of vegetable yields and N_2_O emissions to different management practices in the open vegetable fields is essential to achieve the sustainable vegetable production in China.

The N_2_O in soil is mainly produced via the microbial processes of nitrification and denitrification [16]. The N fertilizer management could regulate the microbially meditated production and emission of N_2_O from soil directly or indirectly by altering the soil conditions, such as N availability and pH [17,18]. Numerous studies indicated that the soil N_2_O emissions were positively correlated with the application rate of N fertilizer, as the surplus N in soil was a major source of reactive N losses [19,20]. Optimizing the N application rate could significantly decrease the N_2_O emissions from agricultural soils [21,22]. Previous studies suggested that reduced fertilizer N rate could mitigate the N_2_O losses from the vegetable cropping systems by 18–57% [23,24].

In addition, applying N fertilizer with the right source offers an effective approach to improve the N use efficiency (NUE) and reduce the fertilizer-induced N_2_O emissions in croplands [25,26]. Enhanced-efficiency N fertilizers (EENFs) (e.g., controlled-release urea fertilizers [CRFs] and inhibitors) have been developed and widely adopted to better synchronize N supply with crop N uptake by retarding the fertilizer N release or microbial N transforming, thereby enhancing NUE and preventing N losses, such as N_2_O [27,28]. The EENF application was estimated to significantly mitigate the N_2_O emissions from vegetable fields by 36% [29]. However, the mitigation effectiveness of EENFs varied with EENF types and management practices [30,31]. Previous meta-analyses reported that the CRFs and nitrification inhibitors (NIs) could significantly reduce the soil N_2_O emissions by 19–74% and 33–58%, respectively [32,33]. Additionally, the surplus N in soil is found to directly impact the EENF effectiveness, and EENFs provide a greater opportunity for reducing fertilizer N inputs and N_2_O losses in high-N-surplus cropping systems [34,35]. Therefore, it is critical to establish an optimal EENF application strategy specifically targeting the Chinese open-field vegetable systems.

China is the leading lettuce (*Lactuca sativa* L.) producer worldwide, with an annual planting area and output of 6.06 × 10^5^ ha and 1.43 × 10^7^ t in 2020, respectively [36]. As one of the most widely planted and consumed leafy vegetables in China, lettuce is rich in essential nutrients, such as vitamin C, vitamin E, lutein and fibers, and plays an important role in promoting health and reducing the risk of numerous chronic diseases [37,38]. However, the environmental risk of lettuce production in North China has been increasing with the increased demand for this nutrient-dense crop, as most local farmers apply excessive fertilizer N to obtain high vegetable yields [39]. In order to determine the agronomically, environmentally and economically optimal EENF application strategy for the open-field lettuce production in China, the impacts of reduced N application combined with different EENFs (i.e., CRF and NI) on the lettuce yield, N_2_O emissions and net economic benefits were comprehensively investigated in this study based on a 4 yr open-field lettuce experiment located in the suburb of Beijing and a cost–benefit analysis (CBA).

## 2. Results and Discussion

### 2.1. Impacts of EENFs on Lettuce Growth and NUE

On average, the lettuce yields under FP, OPT, CU and CUNI treatments were significantly increased by 54.9% in 2017, 144.2% in 2018, 276.1% in 2019 and 646.6% in 2020 compared to CK, without significant differences among them (Table 1). This lack of difference in the vegetable yields between FP and other N fertilization treatments is consistent with prior Chinese open-field vegetable studies [24,29,40], indicating a great potential for fertilizer N input reduction in open vegetable fields without compromising vegetable growth and yield.

Lettuce reached the highest NUE under EENF treatments (CU in 2017 and CUNI in 2018–2020), and a significant increase of 9.1–14.1% and 7.7–8.4% in NUE was found for CU and CUNI relative to FP, respectively (Table 1). Numerous studies observed a similar significant enhancement in NUE (by 6.1–14.9%) when adopting EENFs at a reduced N application rate (by 20–33%) in Chinese open-field leafy vegetable systems compared to conventional N management [41,42,43]. This is expected considering the great effectiveness of EENFs in mitigating reactive N losses in vegetable fields [44,45]. The performance of CUNI for increasing NUE was more consistent than CU in this study (Table 1). This was probably owing to the effective reduction in N_2_O without triggering the tradeoff between NH_3_ and N_2_O emissions, as the N fertilizers were less susceptible to NH_3_ losses when incorporated into soil [35]. Contrary to previously reported findings by Suter et al. [46], the EENF treatments (except for CUNI in 2018) did not further improve the NUE of lettuce system compared to OPT (Table 1), which was probably due to the limited capacity of EENFs for promoting N uptake in this study (Figure 1).

The N uptake of lettuce plants under N fertilization treatments was generally found to be significantly higher than CK, regardless of growing stages, during 2017–2020 (Figure 1). However, similar with vegetable yields, no significant differences were observed in the N uptake among different N fertilization treatments (Figure 1), implying that the lettuce growth was not further promoted by the addition of EENFs relative to reducing N rate alone.

### 2.2. Impacts of EENFs on Soil NH_4_^+^–N and NO_3_^−^–N Contents

Our results showed that the average soil NH_4_^+^–N contents of all treatments declined during the lettuce growing seasons in 2017–2020, and significantly greater NH_4_^+^–N contents were observed for N fertilization treatments compared to CK, despite lettuce growing stage. The average NH_4_^+^–N contents of CK and N fertilization treatments ranged from 0.88 to 2.06 mg kg^–1^ and from 11.14 to 15.99 mg kg^–1^, respectively (Figure 2a). Though FP and OPT were not significantly different, the highest and lowest soil NH_4_^+^–N contents were, respectively, found under the CU and CUNI treatments at the seedling stage (Figure 2a), mainly due to the delayed N release from CRF and the accumulation of NH_4_^+^–N induced by NI via inhibited nitrification [47,48]. The FP, OPT, CU and CUNI treatments did not lead to significantly different average soil NH_4_^+^–N contents at the rosette and mature stages (Figure 2a), probably because NO_3_^−^–N was favored over NH_4_^+^–N by lettuce uptake [49].

In contrast, the average NO_3_^−^–N contents in soil presented a distinctive variation pattern from NH_4_^+^–N in this study. While the average NO_3_^−^–N contents under CK, FP and OPT declined throughout the lettuce growing season, those under CU and CUNI treatments reached the highest level at the rosette stage (Figure 2b). The elevated NO_3_^−^–N under CU and CUNI at the rosette stage probably resulted from the recovered soil nitrification after the EENFs lost effectiveness via biodegradation [50,51]. The average NO_3_^−^–N contents of CK and N fertilization treatments ranged between 2.55 and 5.22, and 27.47 and 39.62 mg kg^–1^, respectively (Figure 2b). In comparison to FP, a reduction of 12.7% in soil NO_3_^−^–N contents for CU and 24.1% for CUNI were observed at the seedling stage (Figure 2b), owing to the inhibited soil N transformation via EENFs when they were most effective [28]. The less evident response of soil NO_3_^−^–N to EENF application at later lettuce growing stages were probably due to the limited effectiveness duration of EENFs, as mentioned earlier (Figure 2b).

### 2.3. Impacts of EENFs on N_2_O Emissions

The soil N_2_O fluxes under N fertilization treatments, including FP, OPT, CU and CUNI, followed a similar pattern during the lettuce growing seasons in 2017–2020. Three N_2_O peaks were found for FP and OPT treatments and one for CU and CUNI after each fertilizer application event (Figure 3), showing that N fertilization was the major driver of soil N_2_O emissions. Precipitation was also found to be a critical regulator for soil N_2_O emissions [52]. However, the relatively low precipitation intensity and frequency during the lettuce growing seasons in this study implied a limited impact of precipitation on soil N_2_O emissions (Figure 4). The peak N_2_O fluxes from CK, FP, OPT, CU and CUNI treatments ranged from 9.1 to 21.7, 40.6 to 56.3, 36.3 to 54.0, 21.0 to 41.4 and 19.5 to 39.9 g N ha^–1^ d^–1^, respectively (Figure 3). The peak N_2_O fluxes of EENF treatments appeared 2–8 days later than the first N_2_O peaks of FP and OPT treatments (Figure 3) due to the delayed urea hydrolysis, retarded nitrification and impeded accumulation of NO_3_^−^ in soil via the addition of CRF and NI [48,53], as supported by the lower soil NH_4_^+^–N and NO_3_^−^–N contents during the early lettuce growing stage in this study (Figure 2) and previous findings [54,55].

The N fertilization treatments resulted in a 115.1–273.3% increase in area-scaled seasonal cumulative N_2_O emissions relative to CK. Compared with FP treatment, OPT, CU and CUNI significantly reduced the area-scaled seasonal cumulative N_2_O emissions by 7.4%, 37.7% and 42.4% on average, respectively. Similarly, the yield-scaled seasonal cumulative N_2_O emissions under OPT, CU and CUNI treatments were significantly lower than FP by 9.6%, 42.4% and 47.5% on average, respectively (Table 2). However, the highly variable yields of CK led to different patterns of yield-scaled emissions across different years in comparison to the N fertilization treatments (Table 1), and the yield-scaled cumulative N_2_O emissions under CK were significantly lower than FP and OPT by 13.1–25.1% and higher than CU and CUNI by 27.9–34.4% on average, respectively (Table 2).

In this study, the adoption of EENFs at an optimized N rate was more effective in mitigating N_2_O emissions than reducing N rate alone. This is as expected, given that the N_2_O generation was further suppressed owing to the decreased accumulation of soil available N via CRF and the impeded nitrification in soil via NI. The CUNI treatment had significantly lower area- and yield-scaled N_2_O emissions than CU in this study. Similar results were reported by Fan et al. [43] and Muller et al. [56], who observed greater reduction in soil N_2_O emissions under the double addition of CRF and NI than the sole amendment of CRF in open vegetable fields. This was mainly attributed to the NI-induced efficient inhibition of soil nitrification and N_2_O production [57]. In addition, the lettuce yields under CUNI tended to be higher than CU (Table 1), thus further lowering the yield-scaled N_2_O emissions.

It was reported that NI amendment could result in higher NH_3_ volatilization through the improved NH_4_^+^–N accumulation in soil and subsequently elevated soil pH [47]. Although the NH_3_ volatilization was not measured in this study, it could be speculated that CUNI treatment did not lead to higher NH_3_ losses, given that soil NH_4_^+^–N availability was a vital modulator of soil NH_3_ emission [58] and there existed no significant differences basically in soil NH_4_^+^–N contents among different N fertilization treatments in this study (Figure 2a). The incorporation of basal N fertilizers into the soil in this study could have prevented the NH_3_ volatilization to a large extent [59]. Additionally, a previous study found that the application of CRF could offset the negative impacts of NI on NH_3_ mitigation [43].

In summary, this study showed that, compared with conventional N fertilizer management, the application of EENFs reduced the N_2_O emissions mainly by decreasing the NO_3_^−^–N contents in soil during the early lettuce growing stage via the retarded N release from CRF and the NI-induced inhibition of microbial nitrification, thereby effectively improving the NUE and alleviating the negative environmental impacts while maintaining high lettuce yield. Similarly, the optimum agronomic and environmental performance of CUNI treatment resulted from the greater reduction in soil NO_3_^−^–N contents and N_2_O loss potential under the double application of CRF and NI.

### 2.4. Impacts of EENFs on Net Economic Benefit of Lettuce Production

The CBA results showed that the N fertilization treatments significantly improved the yield benefits compared to CK, and no significant differences were found among FP, OPT, CU and CUNI treatments. In contrast, OPT, CU and CUNI treatments led to a significant decrease of 8.1%, 38.0% and 42.6% in the N_2_O reduction costs on average compared to FP, respectively. The net economic benefits of lettuce production under OPT, CU and CUNI treatments were higher than that under FP by USD 281, USD 871 and USD 1024 ha^–1^ on average, respectively, among which only CUNI treatment had significantly higher net benefit than FP (by 11.1%) (Table 3).

In this study, the sole optimization of N rate slightly increased the net benefits of a lettuce system compared to local farmers’ practice, showing the potential of decreasing fertilizer N inputs without economic penalty in Chinese open-field vegetable production. In line with previous reports [60,61], applying EENFs at a reduced N rate exceeded the economic performance of optimizing N rate alone, which indicated that the agronomic, environmental and fertilizer- and labor-input saving benefits of EENFs outweighed their extra costs. In addition, it is noteworthy that the implementation of EENFs allowed for lower N fertilizer inputs, thus further reducing the GHG emissions caused by the upstream fertilizer production [62]. The greatest net benefits of lettuce production were derived from the CUNI treatment, implying that the double amendment of CRF and NI combined with N rate optimization could be the most promising approach for allowing the open-field vegetable production in China to achieve high yields as well as great environmental sustainability and economic benefits.

## 3. Materials and Methods

### 3.1. Study Site

The field experiment was carried out in Zhaoquanying Town, Shunyi District in suburban Beijing in North China (116°35′59″ E, 40°13′6″ N), with a temperate semi-humid continental monsoon climate. The average annual air temperature and precipitation (mostly occurring in July and August) are 11.5 °C and 625 mm, respectively. The daily precipitation and mean air temperature during the lettuce growing seasons in 2017–2020 were obtained from a local weather station near the study site (Figure 4). The soil type is classified as a cinnamon fluvo-aquic loam soil. The topsoil (0–20 cm) properties of the study site are as follows: pH 7.89, bulk density 1.31 g cm^–3^, organic matter 1.03%, total N1.10 g kg^–1^, available phosphorus (P) 13.2 mg kg^–1^, and available potassium 136.8 mg kg^–1^.

### 3.2. Experimental Design and Field Management

The treatments include five N fertilization practices: (1) CK: no N fertilizer; (2) FP: conventional urea (46% N) at 255 kg N ha^–1^ based on local farmers’ practices; (3) OPT: conventional urea (46% N) at 204 kg N ha^–1^; (4) CU: polyurethane-coated urea (44% N, 50-day release) at 204 kg N ha^–1^; (5) CUNI: polyurethane-coated urea at 204 kg N ha^–1^ in combination with NI (3,4-dimethyl pyrazole phosphate (DMPP), 1 kg ha^–1^).

The experiment followed a randomized complete block design with four replicates per treatment. The area of each plot was 15 m^2^ (5 m × 3 m). A local head lettuce variety of ‘Sheshou No. 101′ was chosen for the open-field experiment during 2017–2020. The lettuce seeds were sterilized with 50% carbendazim solution (*v*/*v*) for 20 min, and then washed with deionized water four times after being soaked for 8 h each time. Then, the sterilized seeds were wrapped in lint and kept in the incubator at 16 °C for 1 d to promote germination. The lettuce seeds were sown in the seedbeds afterward at a rate of 300 g ha^–1^ and cultivated for 25 days before transplant. The soil of the experimental field was plowed to a depth of 20 cm beforehand with a rotary tiller. The lettuce was transplanted with two rows on each ridge (four ridges per plot) on September 7th in 2017 and 2018, September 8th in 2019 and September 9th in 2020, and was harvested on November 4th in 2017 and 2018, and October 31st in 2019 and 2020.

The 36% of the total N was applied as the basal fertilizer at the seedling stage, 41% as the first top-dressing fertilizer at the rosette stage, and 23% as the second top-dressing fertilizer at the heading stage for both FP and OPT treatments, while all N was applied as the basal fertilizer for CU and CUNI treatments. The P and K fertilizers were applied as basal fertilizers for all treatments at rates of 41 kg P ha^−1^ and 197 kg K ha^−1^. Basal fertilizers were surface applied before lettuce transplant, and then incorporated into soil, whereas top-dressing fertilizers were applied with irrigation. The same experimental arrangement was repeated for four years (2017–2020).

### 3.3. Sampling and Measurements

Five lettuce plants were randomly sampled each time from each experimental plot at the seedling, rosette, heading and mature stages, respectively. The lettuce plants harvested at the mature stage were weighed for fresh yield. The sampled plants were then dried in an oven at 70 °C for 2 days and weighed for dry matter yield. The plant N content was determined via the method of Kjeldahl digestion [63]. The plant N uptake was calculated as follows:
(1)$$N_{\mathrm{uptake}}=N_{\mathrm{plant}}\times {\mathrm{DM}}_{\mathrm{plant}},$$
where N_uptake_ is the plant N uptake (kg N ha^–1^), N_plant_ is the plant N content (%), and DM_plant_ is the lettuce dry matter yield (kg ha^–1^). The NUE was obtained by the following equation:
(2)$$\mathrm{NUE}=\frac{N_{F}-N_{C}}{N_{\mathrm{rate}}},$$
where NUE is the N use efficiency (%), N_F_ and N_C_ are the plant N uptake (kg N ha^–1^) with and without N fertilization, and N_rate_ is the application rate of fertilizer N (kg N ha^–1^) under N fertilization treatments.

The ring knife method was adopted to measure the soil bulk density in situ [64]. Five soil samples (0–20 cm) were randomly collected each time from each experimental plot at the seedling, rosette and mature stages, respectively, and then homogeneously mixed into a composite sample for chemical analysis. The sieved soil was made into a slurry by adding deionized water with a soil-to-water ratio of 1:5 (*w*/*v*), and then measured by a pH meter for soil pH. The organic matter content in soil was determined using the Walkley & Black method [65]. The total N content in soil was measured by the dry combustion method [66]. Soil NH_4_^+^–N and NO_3_^−^–N were extracted with 1.0 M KCl solution and then measured for NH_4_^+^–N and NO_3_^−^–N contents with a continuous flow injection analyzer [67].

Soil N_2_O emissions were determined by the static chamber-gas chromatography method [68]. Polyvinyl chloride plastic-made column chambers (34.5 cm in diameter, 60 cm high) installed with a digital thermometer, an electric fan and a vent tube inside were used for N_2_O sampling. Two chamber bases per experimental plot were inserted 10 cm deep into the soil after lettuce transplant, covering one lettuce plant under each chamber during the gas sampling. The samplings were carried out between 9:00 am and 11:00 am every day following each N fertilization or precipitation event for 3–5 days or once every 5–10 days otherwise. Four gas samples were successively collected from the chamber headspace in each plot at an interval of 10 min with a 20 mL gas-tight syringe, and then immediately transferred to 12 mL air-evacuated gas-tight glass vials. The N_2_O concentrations of gas samples were measured by a gas chromatography (Shimadzu, GC-14B, Japan). The N_2_O flux was obtained via the equation below [68]:
(3)$$F=H\times \rho \times \left(\frac{\Delta c}{\Delta t}\right)\times \left(\frac{273}{273+T}\right),$$
where F is the N_2_O flux (mg m^–2^ h^–1^), H is the height of the chamber headspace (m), ρ is the density of N_2_O standard gas (mg m^–3^), Δc/Δt is the changing rate of gas concentration in the chamber (mg m^–3^ h^–1^), and T is the chamber mean temperature during the gas sampling (°C). The area-scaled seasonal cumulative N_2_O emissions were calculated as follows [68]:
(4)$$\mathrm{ASE}=\sum_{i}^{n}(\frac{\left(F_{i}+F_{i+1}\right)\times 24}{2\times 1000}\times D_{i})\times \frac{14}{44}\times 10,$$
where ASE is the area-scaled seasonal cumulative N_2_O emission (kg N_2_O-N ha^–1^ season^–1^), F_i_ and F_i+1_ are the N_2_O fluxes measured on two adjacent sampling dates (mg m^−2^ h^−1^), D_i_ is the length of the ith sampling interval (d), n is the total number of sampling intervals, and 44 and 14 are the relative molecular masses of N_2_O and N, respectively. The yield-scaled seasonal cumulative N_2_O emissions were derived from dividing ASE by lettuce yield.

### 3.4. CBA Analysis

The net economic benefits of the lettuce production system under different N fertilization practices were estimated by deducting the costs of N fertilizer products (i.e., conventional urea and EENFs), N_2_O-induced environmental impacts and labor inputs for N fertilization from the benefits of lettuce yield. The local market prices of USD 0.14 kg^–1^, USD 352 t^–1^, USD 487 t^–1^ and USD 21 kg^–1^ were adopted for lettuce, conventional and polyurethane-coated urea, and DMPP, respectively [69]. The carbon price of USD 7.38 t^–1^ CO_2_-equivalent was used to assess the N_2_O reduction costs [70]. Four persons were employed for basal fertilization and two persons for top-dressing fertilization, with a payment of USD 28 per person.

### 3.5. Statistical Analysis

Statistical analysis was carried out with SPSS (SPSS Inc., Chicago, IL, USA; Version 21.0). The Shapiro–Wilk and Levene tests were used to verify the normality and homogeneity of variance before the analysis of variance (ANOVA). One-way and two-way repeated measures ANOVA were adopted to analyze the data obtained at the mature stage and at different lettuce growing stages, respectively, which was followed by the Tukey honest significant difference test if the treatment effects were significant. All levels of significance were defined at *p* ≤ 0.05. All plots were illustrated using Excel (Microsoft Corporation, Redmond, WA, USA; Version Microsoft 365).

## 4. Conclusions

The adoption of EENFs (CRF and NI) significantly decreased the seasonal cumulative N_2_O emissions while maintaining high lettuce yield and reducing fertilizer N inputs compared with conventional fertilization. The double amendment of CRF and NI led to the lowest N_2_O emissions and the highest NUE as a result of the effectively inhibited soil NO_3_^−^–N accumulation. In addition, CBA revealed that the yield-boosting, environment-preserving and fertilizer- and labor-input saving benefits of EENFs outweighed their extra costs, thus improving the net benefits of the lettuce production system. Overall, our study highlighted that the double addition of CRF and NI combined with N rate optimization would be the optimal N fertilization practice to reach the environmentally and economically sustainable production of open-field vegetables in China, especially when concurrently employing other 4R practices, such as fertilizer incorporation to prevent the soil NH_3_ volatilization.

## Figures and Tables

**Figure 1 plants-12-00081-f001:**
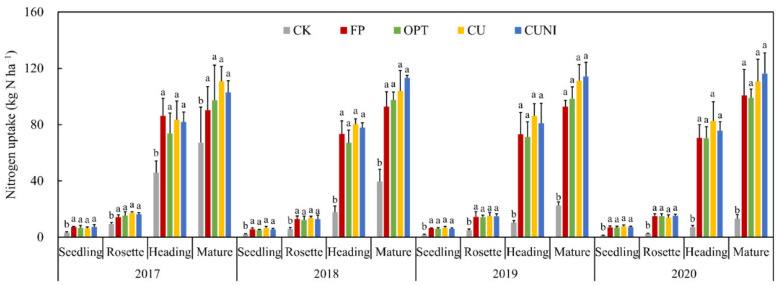
Nitrogen (N) uptake during lettuce growing seasons in 2017–2020. CK, no N fertilizer; FP, conventional urea based on local farmers’ practices; OPT, conventional urea at reduced N rate; CU, polyurethane-coated urea at reduced N rate; CUNI, polyurethane-coated urea at reduced N rate in combination with nitrification inhibitor (NI). Same letter indicates no significant differences at the level of 0.05. Bars represent standard deviations (*n* = 4).

**Figure 2 plants-12-00081-f002:**
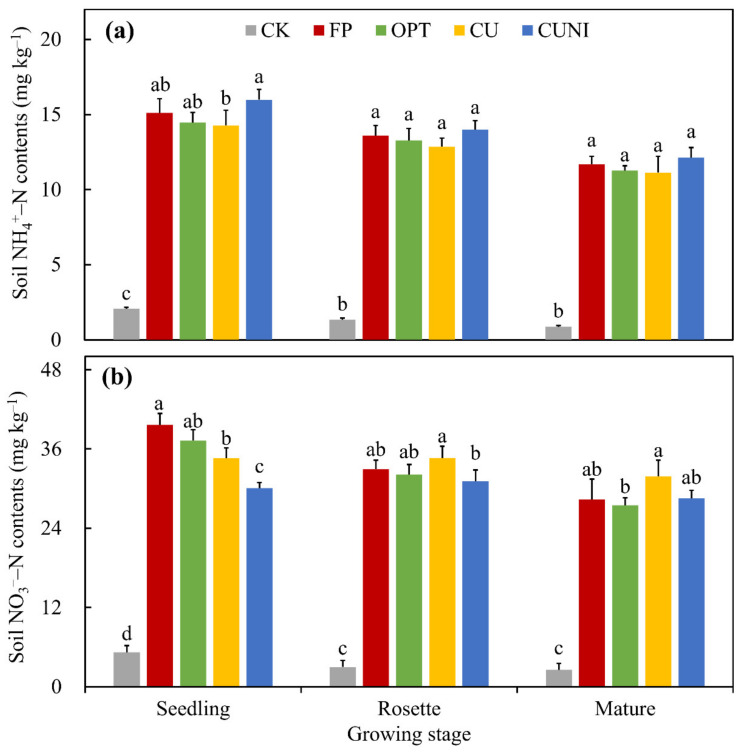
Average soil NH_4_^+^–N (**a**) and NO_3_^−^–N (**b**) contents during lettuce growing seasons in 2017–2020. CK, no N fertilizer; FP, conventional urea based on local farmers’ practices; OPT, conventional urea at reduced N rate; CU, polyurethane-coated urea at reduced N rate; CUNI, polyurethane-coated urea at reduced N rate in combination with NI. Same letter indicates no significant differences at the level of 0.05. Bars represent standard deviations (*n* = 4).

**Figure 3 plants-12-00081-f003:**
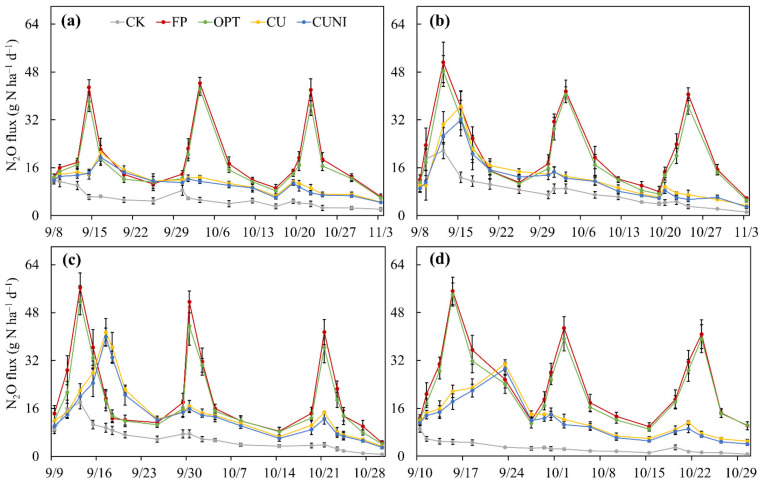
Soil N_2_O fluxes during the lettuce growing seasons in 2017 (**a**), 2018 (**b**), 2019 (**c**) and 2020 (**d**). CK, no N fertilizer; FP, conventional urea based on local farmers’ practices; OPT, conventional urea at reduced N rate; CU, polyurethane-coated urea at reduced N rate; CUNI, polyurethane-coated urea at reduced N rate in combination with NI. Bars represent standard deviations (*n* = 4).

**Figure 4 plants-12-00081-f004:**
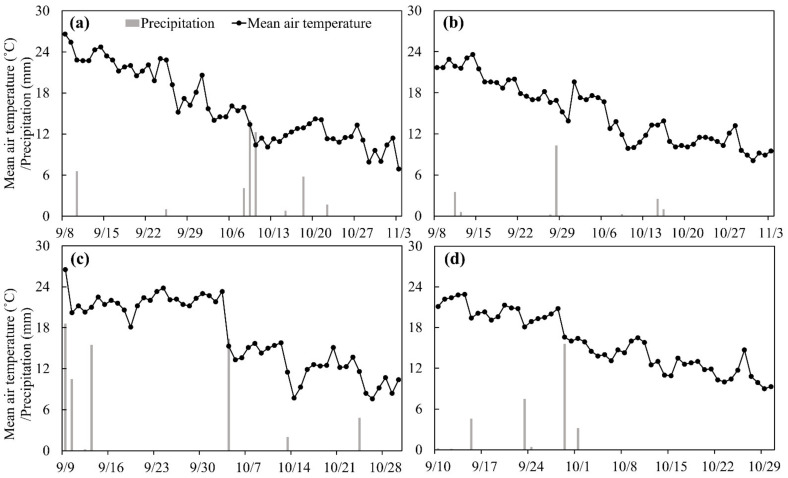
Daily precipitation and mean air temperature during the lettuce growing seasons in 2017 (**a**), 2018 (**b**), 2019 (**c**) and 2020 (**d**).

**Table 1 plants-12-00081-t001:** Lettuce yield and N use efficiency (NUE) under different treatments during 2017–2020.

Treatment	2017	2018	2019	2020	Avg.
Lettuce yield (Mg ha^–1^)					
CK	46.3 (11.3) ^1^ b	30.0 (5.8) b ^2^	19.3 (1.4) b	9.8 (1.5) b	26.4 (3.8) b
FP	66.2 (7.2) a	70.4 (4.4) a	69.4 (5.3) a	70.4 (7.6) a	69.1 (2.6) a
OPT	70.9 (11.4) a	73.4 (4.7) a	69.3 (6.3) a	69.7 (4.9) a	70.8 (5.0) a
CU	74.0 (9.5) a	74.4 (8.8) a	74.5 (5.5) a	75.9 (5.1) a	74.7 (6.6) a
CUNI	75.9 (9.8) a	75.1 (7.6) a	77.3 (7.2) a	75.6 (8.5) a	76.0 (2.3) a
NUE (%)					
FP	9.9 (7.4) b	22.5 (6.9) c	29.4 (5.5) b	37.1 (12.7) b	24.7 (7.4) b
OPT	14.8 (12.2) ab	28.4 (2.8) bc	37.3 (4.1) ab	42.1 (3.0) ab	30.6 (3.5) ab
CU	21.3 (5.1) a	31.6 (7.2) ab	43.5 (5.6) a	47.8 (7.8) ab	36.1 (6.1) a
CUNI	17.5 (4.1) ab	36.1 (0.9) a	45.0 (4.9) a	50.6 (7.2) a	37.3 (2.3) a
*F* value and probability level					
Lettuce yield					
Treatment	5.8 **	36.0 ***	76.1 ***	88.4 ***	92.4 ***
NUE					
Treatment	5.5 **	37.7 ***	66.3 ***	29.6 ***	42.9 ***

^1^ Values in brackets are standard deviations (*n* = 4). ^2^ Same letter within column means no significant difference at the level of 0.05. ** Significant at *p* < 0.01; *** significant at *p* < 0.001. CK, no N fertilizer; FP, conventional urea based on local farmers’ practices; OPT, conventional urea at reduced N rate; CU, polyurethane-coated urea at reduced N rate; CUNI, polyurethane-coated urea at reduced N rate in combination with NI.

**Table 2 plants-12-00081-t002:** Area- and yield-scaled seasonal cumulative N_2_O emissions under different treatments during 2017–2020.

Treatment	2017	2018	2019	2020	Avg.
Area-scaled cumulative emissions (kg N ha^–1^ season^–1^)					
CK	0.30 (0.01) ^1^ e	0.46 (0.02) e ^2^	0.31 (0.01) e	0.14 (0.004) e	0.30 (0.01) e
FP	1.05 (0.02) a	1.24 (0.05) a	0.98 (0.03) a	1.22 (0.03) a	1.12 (0.02) a
OPT	0.96 (0.03) b	1.13 (0.05) b	0.92 (0.04) b	1.14 (0.03) b	1.04 (0.01) b
CU	0.64 (0.01) c	0.74 (0.02) c	0.74 (0.02) c	0.67 (0.03) c	0.70 (0.01) c
CUNI	0.60 (0.01) d	0.68 (0.01) d	0.68 (0.02) d	0.62 (0.03) d	0.65 (0.01) d
Yield-scaled cumulative emissions (g N Mg^–1^ season^–1^)					
CK	6.38 (0.22) e	15.2 (0.64) b	16.1 (0.46) a	14.1 (0.39) c	13.0 (0.36) c
FP	15.8 (0.28) a	17.6 (0.65) a	14.0 (0.43) b	17.3 (0.44) a	16.2 (0.32) a
OPT	13.6 (0.37) b	15.5 (0.61) b	13.2 (0.56) c	16.3 (0.47) b	14.6 (0.11) b
CU	8.61 (0.15) c	9.95 (0.24) c	9.90 (0.26) d	8.87 (0.39) d	9.33 (0.17) d
CUNI	7.94 (0.17) d	9.09 (0.14) d	8.81 (0.24) e	8.15 (0.33) e	8.50 (0.13) e
*F* value and probability level					
Area-scaled cumulative emissions					
Treatment	1291 ***	436.9 ***	426.0 ***	1068 ***	2410 ***
Yield-scaled cumulative emissions					
Treatment	1033 ***	217.9 ***	217.0 ***	428 ***	769.5 ***

^1^ Values in brackets are standard deviations (*n* = 4). ^2^ Same letter within column means no significant difference between treatments at the level of 0.05. *** Significant at *p* < 0.001. CK, no N fertilizer; FP, conventional urea based on local farmers’ practices; OPT, conventional urea at reduced N rate; CU, polyurethane-coated urea at reduced N rate; CUNI, polyurethane-coated urea at reduced N rate in combination with NI.

**Table 3 plants-12-00081-t003:** The average benefits of lettuce yield (USD ha^–1^) and costs of purchasing N fertilizers, reducing N_2_O-induced environmental impacts and N fertilization-related labor inputs (USD ha^–1^) under different treatments during 2017–2020.

Treatment	Yield Benefit	Fertilizer Cost ^1^	Environmental Cost of N_2_O	Labor Cost ^1^	Net Benefit
CK	3690 (533) ^2^ b	0	2.09 (0.06) e	0	3688 (533) c
FP	9675 (362) a ^3^	195.1	7.82 (0.14) a	224	9248 (362) b
OPT	9916 (697) a	156.1	7.19 (0.06) b	224	9528 (697) ab
CU	10,461 (919) a	225.8	4.85 (0.09) c	112	10,118 (919) ab
CUNI	10,635 (323) a	246.8	4.49 (0.07) d	112	10,272 (323) a
*F* value and probability level					
Treatment	92.4 ***	-	2724 ***	-	82.4 ***

^1^ Fertilizer and labor costs for each treatment are constant values. ^2^ Values in brackets are standard deviations (*n* = 4). ^3^ Same letter within column means no significant difference at the level of 0.05. *** Significant at *p* < 0.001. CK, no N fertilizer; FP, conventional urea based on local farmers’ practices; OPT, conventional urea at reduced N rate; CU, polyurethane-coated urea at reduced N rate; CUNI, polyurethane-coated urea at reduced N rate in combination with NI.

## Data Availability

The data presented in this study are available on request from the corresponding author.

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
