# Peer review of "Enhanced-Efficiency Fertilizers Impact on Nitrogen Use Efficiency and Nitrous Oxide Emissions from an Open-Field Vegetable System in North China"

_plants, 2022, doi:10.3390/plants12010081_

Round 1
Reviewer 1 Report
The research idea shows a good degree of novelty, with particular regard to the application Enhanced-efficiency fertilizers impact on nitrogen use efficiency and nitrous oxide emissions from an open-field vegetable system in North China. The manuscript is readable, organized and appropriate length, according to data presented. The topic discussed in logical sequence. The content of the manuscript is clear, understandable without any unnecessary repetition. This paper needs some fixes
In line 6 delete “and”
Abstract
Line 16 treatment “vi” must become “v”
Introduction
The introduction describes quite well the main aspects treated by this study.
Lines 28-37: I suggest authors include more recent data and references.
Line 99-105: I suggest the authors to put in evidence and emphasis on the purpose of the research, not be limited to describe the evidence
The material and methods chapter goes after the results, as per the Plant template
Materials and methods
Line 135-137: rewriting the sentence it is not clear the percentages to whom they refer
Line 146: I suggest "dry matter" to "dry mass"
Had not the authors a weather station?? Since the nitrous oxide fluxes are greatly influenced by the climate, I suggest inserting a paragraph on the thermopluviometric trend of the 4-year test.
Furthermore, if it has been monitored, I would also include the soil moisture trend over the 4 years
Results
Line214-216 : The authors state that the yields are significantly higher than CK but report only the percentage increase of CUNI and FP. Recommend to bring back or all percentage increases or an average increase between 4 significantly different treatments.
Line 219-221: as before
Line 223: A discussion on increasing the dry mass is missing
Line235: I suggest the authors broaden the discussion of fluxes, especially for inhibitor fertilizer
Line374: decrease the character “open-field vegetable production in China.”
Reviewer 2 Report
The idea of the investigation is clearly presented. The research topic remains relevant in worldwide scale. I have almost no remarks regarding Introduction, Materials and methodology.
Results and Discussion chapter. I would like suggest you at first describe table 1 and latter Figure 1. I am not sure if lettuce DM as parameter is very important parameter. You may easily remove Fig.1 (a)?
I lacked any discussion at the end of this chapter. You did much work when investigating many parameters in your work. Hope you write a brief summary discussion by combining all the parameters studied.
Round 2
Reviewer 1 Report
Dear authors,
Congratulations on the revision work done.
Accept in present form
Author Response
Dear Reviewer,
Thank you very much for your comments.
Best regards,
Wentian He
Reviewer 2 Report
I would like to thank the authors for the improving the text of the manuscript. I think the manuscript should be published in Plants magazine.
Author Response

(The authors gave the same response as above.)
